# Personalized Medicine for TP53 Mutated Myelodysplastic Syndromes and Acute Myeloid Leukemia

**DOI:** 10.3390/ijms221810105

**Published:** 2021-09-18

**Authors:** Thomas Cluzeau, Michael Loschi, Pierre Fenaux, Rami Komrokji, David A. Sallman

**Affiliations:** 1Hematology Department, University Hospital of Nice, Cote d’Azur University, 06200 Nice, France; loschi.m@chu-nice.fr; 2INSERM U1065, Mediterranean Center of Molecular Medicine, Cote d’Azur University, 06200 Nice, France; 3French Group of Myelodysplasia, 75010 Paris, France; pierre.fenaux@aphp.fr; 4Senior Hematology Department, Saint Louis Hospital, Paris 7 University, 75010 Paris, France; 5Moffit Cancer Center and Research Institute, Tampa, FL 33612, USA; Rami.Komrokji@moffitt.org (R.K.); David.Sallman@moffitt.org (D.A.S.)

**Keywords:** MDS, AML, TP53, magrolimab, eprenetapopt

## Abstract

Targeting TP53 mutated myelodysplastic syndromes and acute myeloid leukemia remains a significant unmet need. Recently, new drugs have attempted to improve the outcomes of this poor molecular subgroup. The aim of this article is to review all the current knowledge using active agents including hypomethylating agents with venetoclax, eprenetapopt or magrolimab. We include comprehensive analysis of clinical trials to date evaluating these drugs in TP53 myeloid neoplasms as well as discuss future novel combinations for consideration. Additionally, further understanding of the unique clinicopathologic components of TP53 mutant myeloid neoplasms versus wild-type is critical to guide future study. Importantly, the clinical trajectory of patients is uniquely tied with the clonal burden of TP53, which enables serial TP53 variant allele frequency analysis to be a critical early biomarker in investigational studies. Together, significant optimism is now possible for improving outcomes in this patient population.

## 1. Introduction

Of all somatic mutations identified in patients with myelodysplastic syndromes (MDS) and acute myeloid leukemia (AML), TP53 mutations are associated with the most inferior outcomes across independent studies, with a median OS of 6–12 months [1,2,3]. Additionally, TP53 mutations predict inferior OS for all standard of care therapies, including allogeneic hematopoietic stem cell transplantation (HSCT), which represents the only curative option for these patients [4,5,6,7,8,9,10]. TP53 protein has an impact on several cellular functions as cancer, aging, senescence and DNA repair, but its role in AML remains enigmatic. TP53 inactivation by mutation or deletion enhances the effect of other oncogenes and promotes the proliferation of cancer cells [11]. Notably, *TP53* mutations impart intrinsic resistance to cytotoxic chemotherapies and thus, the development of novel therapies requires their activity independent of wild-type p53 function or requires the restoration of wild-type function. In myeloid malignancies utilizing CRISPR-Cas9 to generate isogenic human leukemia cell lines of the most common *TP53* missense mutations, Boettcher and colleagues identified that *TP53* missense mutations result in a dominant-negative effect, ultimately leading to a selection advantage when exposed to DNA damage [12]. Evolutionary Action score (EAp53) is a computationally derived score to quantify the deleterious impact of different missense *TP53* mutations, with higher scores representing the most negative impact on p53 function. Importantly, a high EAp53 was an independent negative prognostic covariate for outcomes in *TP53* mutated MDS/AML. Thus, particularly for mutations with high EAp53 scores, agents to activate wild-type p53 function are of high importance, whereas the few patients with low EAp53 scores (~6% of patients) could potentially still utilize strategies that depend on residual p53 function. Importantly, the TP53 variant allele frequency (VAF) or allelic state are tightly concordant with outcomes in this molecular subgroup [1,13,14,15]. In both MDS and AML patients, a VAF > 40% is a strong prediction of inferior overall survival (OS). Notably, the only TP53 patients that may have a more indolent course would be monoallelic patients who have a VAF < 20% and are non-complex, as occurs in the setting of isolated deletion 5q patients [16]. Unfortunately, the vast majority of higher-risk MDS (HR-MDS) and AML patients have adverse prognostic features in regard to the TP53 mutation (~>80–85%). Importantly, the VAF of TP53 can be serially followed and is intimately correlated with the clinical trajectory of the patient [17]. Specifically, expansion of TP53 VAF is associated with poor OS and is often directly tied to progression/relapse, or in some cases, may forecast a future progression. Similarly, clearance of VAF, even a VAF < 5%, was predictive of improved OS. From a treatment perspective, clearance of TP53 with hypomethylating agent (HMA) prior to allo-HSCT was strongly concordant with favorable outcome, whereas patients who did not have clearance appeared to have no benefit from allo-HSCT, thus serving as a critical marker for which patients should proceed to allo-HSCT and also strongly supporting that serial TP53 should be followed for investigational therapies. Together, these data highlight the profound negative connotation of *TP53* mutation in MDS/AML and the urgent need for effective, biologically rational, targeted therapies. In this review, we focus on outcomes of standard therapy in TP53 mutant myeloid neoplasms, evaluate investigational therapies which have shown promise in this molecular subset, as well as provide insight into future novel combinations that will ideally finally change the natural history of this challenging disease.

## 2. Treatment Overview with Azacitidine + Venetoclax

Azacitidine (AZA) represents the standard of care of high-risk MDS: median overall survival (OS) was 24.5 months and the overall response rate (ORR) was 29%, including 17% of patients experiencing complete remission (CR) [18]. AZA is a hypomethylating agent and has an impact on the aberrant methylation of CpG islands in the promoter region of several genes observed in MDS and AML [19]. In TP53 mutated MDS, OS was negatively influenced by the presence of TP53 mutation (median OS 12.4 months versus 23.7 months, *p* < 0.0001 and HR 2.89 (95% confidence interval 1.38–6.04; *p* = 0.005). The ORR was 44%, including 22% of patients with CR [20]. Venetoclax (VEN), a selective small-molecule BCL2 inhibitor, induces apoptosis in malignant cells that are dependent on BCL2 for survival. AZA induces a synergistic effect through the down-regulation of MCL1 and the induction of expression of pro-death proteins NOXA and PUMA, enhancing a dependance of leukemia cells to BCL2 [21,22]. The combination AZA + venetoclax (VEN) is under investigation in MDS. Preliminary results of 73 patients from a phase 1 clinical trial were presented at an ASH meeting [23] and showed promising results, with a median OS not reached with a median follow-up at 16.4 months, and the ORR was 79%, including 39.7% of patients with CR and 39.7% with marrow CR (mCR). The safety schedule was AZA 75 mg/m²/d subcutaneously day 1–7 + VEN 400 mg/d orally day 1–14 on 28-day cycles. No data were available in the specific TP53 mutated MDS patients for the moment, although will likely be updated at future presentations. Additionally, a randomized phase 3 clinical trial is ongoing (clinicaltrials.gov NCT: NCT04401748).

AZA has also been the standard of care for patients with AML ineligible for intensive chemotherapy (IC): median OS was 10.4 months and the ORR was 31%, including 28% of patients with CR or CR with incomplete blood count recovery (CRi) [24]. Recently, AZA combined with VEN was approved by the Food and Drug Administration (FDA) and the European Medicines Agency (EMEA) and has become the new standard of care for AML patients ineligible for IC [25]. The combination treatment showed a significant increase in median OS at 14.7 months and the ORR was also significantly higher at 66.4%, including 36.7% of patients with CR and 29.7% with CRi. The schedule was different than in MDS, with AZA 75 mg/m²/d subcutaneously day 1–7 + VEN 400 mg/d orally day 1–28 on 28-day cycles. The safety profile showed an increased but manageable hematological toxicity, with 84% of patients developing infections. Nevertheless, early death mortality was not higher, with death reported for only 7% of patients at day 30. In TP53 mutated AML patients (n = 52), an increase in ORR was observed (55% vs. 0% in patients treated by AZA + VEN vs. AZA + placebo, respectively). In spite of this better ORR, no benefit in OS was observed in this poor prognostic subgroup compared to AZA + placebo. Median OS was 6 months independently of treatment arm and this is in line with other retrospective cohorts of TP53 mutant AML patients treated with hypomethylating agents (HMA) + VEN despite extended decitabine courses with a median OS of ~6 months [25,26]. These data are in correlation with recent prospective data where no improvements in response rate or OS were seen in 10-day versus 5-day decitabine treatments, with over 30% of the cohort representing TP53 mutant patients [27]. To conclude, AZA + VEN became the standard of care for AML ineligible for IC, but the benefit of median OS was not observed in all subgroups of patients, especially in TP53 mutated patients. These data build on recent translational data identifying TP53 mutation to be a direct driver of VEN resistance [28,29]. Together, these data highlight that unfortunately, AZA + VEN has not provided a significant improvement for patients with TP53 mutation and thus, novel therapeutic investigation is warranted, including in comparison to AZA + VEN.

## 3. Treatment Overview with Azacitidine + Eprenetapopt

Eprenetapopt (APR-246) is a novel, first-in-class, small molecule that selectively induces apoptosis in TP53-mutant cancer cells. Eprenetapopt is converted to methylene quinuclidinone (MQ), which covalently binds mutant p53 to restore wild-type conformation, resulting in cell cycle arrest and apoptosis [30]. The mechanism of action is not limited only to the reactivation of p53 pathway, which may have a major impact on the positive effect of eprenetapopt [31]. First, eprenetapopt depletes glutathione (GSH) and increases oxidative stress, inducing reactive oxygen species (ROS) via inhibition of thioredoxin reductase, thioredoxin and glutaredoxin [32,33,34]. Second, eprenetapopt induced ferroptosis that is an important iron-dependent, non-apoptotic programmed cell death pathway characterized by lipid peroxidation [35]. A synergistic effect with AZA was described using TP53 mutated cell lines (both missense and knockout models), mice models and bone marrow samples from TP53 mutated patients [36].

Based on these preclinical results, two clinical trials evaluating the combination of AZA + eprenetapopt were conducted, one in the United States (US) and another in France [37]. The US trial was a phase 1/2 study including intermediate, high and very-high IPSS-R risk MDS and AML with low blast counts (20–30% of blasts) [38]. The schedule of treatment was APR-246 4500 mg/d intravenously over 6 h days 1–4 with AZA 75 mg/m^2^ subcutaneously daily days 4–10 on 28-day cycles. Allogeneic stem cell transplantation (ASCT) was permitted if patients were eligible. Fifty-five patients with a median age of 66 years were included in this study, including 40 MDS patients and 11 AML patients. The ORR was 71%, including 44% of patients with CR and 38% with TP53 NGS negativity. The median duration of CR was 7.3 months, and 40% of included patients proceeded to ASCT. With a median follow-up of 10.5 months, median OS was 10.8 months. The safety profile showed no increased hematological toxicity compared to the historical safety of AZA alone. The French trial was similar except for the inclusion population, the possibility to perform maintenance treatment after ASCT and the characteristics of the population. Fifty-two patients with a median age of 74 years, higher than in the US trial, were included in this phase 2 study. The population was composed of 34 MDS patients and 18 AML patients, including 7 with AML with more than 30% of blasts. The schedule of treatment was the same except for the maintenance phase post-ASCT, where a reduced schedule was used: APR-246 3700 mg/d intravenously over 6 h days 1–4 with AZA 36 mg/m^2^ subcutaneously daily days 1–5 on 28-day cycles The ORR was 58%, including 37% of patients with CR and 30% with TP53 NGS negativity. The median duration of CR was 11.7 months, and 8% of included patients proceed to ASCT. With a median follow-up of 9.7 months, median OS was 12.1 months. The safety profile was similar for the hematological toxicity, but neurological side effects were reported in this trial for 40% of patients (including only 6% of grade 3 or 4). Neurological side effects were fully reversible, and no recurrence was observed after dose reduction. It was significantly related to a lower glomerular filtration rate at treatment onset and a higher age of the included population. 

In these studies, results seemed better in TP53 mutated MDS patients. The ORRs were 62–73%, including 47–50% of patients with CR and 47–48% with TP53 NGS negativity. Based on these results, a phase 3 clinical trial is ongoing to evaluate efficacy of AZA + eprenetapopt in TP53 MDS patients versus azacitidine monotherapy (clinicaltrials.gov NCT: NCT03745716). Notably, according to a recent press release, although the CR rate was higher in the combination arm, this did not reach statistical significance (33.3% vs. 22.4%; *p* = 0.13). Although disappointing, key questions remain to be answered from this study, particularly regarding potential reasons of lower activity in than in the earlier phase 2 trials, potentially via undertreatment in the experimental arm, as well as if subgroups of patients had differential responses. Investigations in AML are ongoing with triplet strategies. In the above studies, the ORR was lower for AML patients (especially in AML patients with more than 30% of blasts). Specifically, the ORR was 45–64%, including 27–36% CR in AML patients with low blast counts, and 14% including 0% CR for AML patients with more than 30% of blasts. Based on these results, a phase 1 study is ongoing evaluating AZA + VEN + eprenetapopt in TP53 mutated AML (clinicaltrials.gov NCT: NCT04214860). Importantly, a maintenance trial with azacitidine is ongoing post allogeneic HSCT given the very high rate of relapse in this molecular subgroup (clinicaltrials.gov NCT: NCT03931291). Finally, an oral form of eprenetapopt (APR-548) combined to AZA is in development in TP53 mutated MDS/AML and could potentially replace eprenetapopt intravenously in the future (clinicaltrials.gov NCT: NCT04638309). Eprenetapopt is a safe and effective targeted therapy in TP53 myeloid neoplasms. The best combination to obtain improved OS and ORR still needs to be evaluated. Furthermore, evaluation of novel combinations in both the frontline and R/R setting are warranted.

## 4. Treatment Overview with Azacitidine + Magrolimab

CD47 is broadly expressed and is the dominant macrophage checkpoint which acts as a “don’t eat me signal” via interaction with signal regulatory protein α (SIPRα) on macrophage, and has been recently reviewed [39]. For macrophage mediated phagocytosis to occur, a “pro-eat me” signal is required and calreticulin is the most well characterized, although others exist [40]. Importantly, pro-eat signals are minimally expressed on normal cells, thus allowing for selective targeting in the setting of malignancy. CD47 expression is increased on cancer cells with higher expression associated with inferior OS, including in AML patients [41,42]. Importantly, these data have also been shown in MDS patients, with higher-risk patients having both increased CD47 expression as well as an increase in the pro-eat me signal calreticulin, which is the dominant pro-phagocytic signal on human cancer cells [40,43]. Given these data, targeting the CD47/SIRPα axis has received strong enthusiasm that has recently exploded in the setting of positive clinical trial data. In this regard, magrolimab, an IgG4 monoclonal antibody to activate antibody dependent cellular phagocytosis, was the first CD47 targeted agent to enter the space in 2014 where monotherapy investigation was performed in R/R AML patients [44]. No MTD was reached in this study and notably, infiltration of CD3+ T-cells was observed, supporting pre-clinical data wherein targeting CD47 can lead to adaptive immune activation. Although anti-leukemia activity has been observed with single agent therapy with magrolimab, mechanistical combinations with agents that can tip the balance of pro-eat me signals are likely required. In support of this hypothesis, preclinical data have strongly supported synergy of azacitidine with the CD47 inhibitor magrolimab [29]. Specifically, the synergy is mediated via upregulation of the pro-eat me signal calreticulin in the setting of azacitidine and in an aggressive AML xenograft model. Combination therapy led to increased macrophage-mediated phagocytosis, disease eradication and improved OS versus with either agent alone, and served as the pre-clinical data supporting this combination in MDS/AML patients. More importantly, these data have been demonstrated clinically with multiple presentations from 2019–2020. At the 2020 EHA Congress, very high response rates were observed in MDS patients (91% ORR, 42% CR), including high responses in TP53 mutant MDS patients (NCT03248479). These promising results supported the FDA Breakthrough Designation and support the ongoing, randomized double-blind placebo-controlled phase 3 ENHANCE study (NCT04313881). More recently, data were presented on the TP53 mutant AML cohort at ASH of 2020. Importantly, in the TP53 mutant AML cohort of evaluable patients (n = 29), the CR/CRi rate was 59% with a median OS of 12.9 months to date, although follow-up was short (median follow-up of 4.7 months). Importantly, patients achieved a high depth of response with 44% complete cytogenetic response and 29% MRD negativity by high sensitivity multi-parameter flow cytometry. These data support the planned phase 3 open-label study of AZA + magrolimab vs. AZA + VEN in unfit AML patients and versus induction chemotherapy in fit patients with a primary endpoint of OS in the non-intensive group (ENHANCE-2; NCT04778397). 

As discussed above, any agent that tips the balance of pro-eat me signals could serve as a combination partner for CD47-SIRPα inhibitor therapies, which could include traditional cytotoxic chemotherapies, venetoclax and others. Additionally, these agents can also synergize with other monoclonal antibodies targeting surface antigen of cancer cells, whereby these agents serve as pro-eat me signals (i.e., acting as an extrinsic pro-phagocytic signal through Fc receptor/macrophage-mediated antibody dependent cellular phagocytosis as occurs with rituximab in B-cell malignancies). Given the potential cross-activation of the adaptive immune response, combination with traditional immune checkpoint inhibitors (e.g., PD-1 or PDL1 inhibitors) is a novel therapeutic option and a trial is ongoing in R/R AML with the anti-PDL1 inhibitor atezolizumab (NCT03922477). As a primary mechanism of action of CD47 targeted therapy may be via leukemia stem cell (LSC) elimination, combining it with other antibody and cellular therapies that also focus on LSC eradication would be of clinical importance. As an example, combination therapy with a TIM3 inhibitor (e.g., sabatolimab) is also warranted. Lastly, evaluating the combination of CD47 inhibitors with other agents with robust activity in MDS/AML will be of keen interest as the frontline treatment paradigm of these patients may rapidly evolve. This would be of particular interest with eprenetapopt in TP53 mutant MDS/AML as well as with pevonedistat [37,38,45]. There are multiple other CD47 monoclonal antibodies in clinical development. Some potential differentiating features may be less anemia, based on differential binding to RBCs, although even with magrolimab, this has clinically not been a major issue via utilization of a priming strategy, as well as the fact that younger RBCs shed CD47 via a process known as RBC pruning. Whether there are key efficacy differences based on specific CD47 monoclonal antibody remains an open question for future study. In addition to CD47 monoclonal antibodies, an additional MOA includes SIRPα fusion proteins (e.g., TTI-621 [IgG1]/TTI-622 [IgG4]/ALX0048, bispecific antibodies, anti-SIRPα antibodies and bi-functional fusion proteins (SL-172154 [SIRPα-Fc-CD40L]), among others [39]). Future clinical data are required to see if there are potentially differential safety or efficacy concerns with agents of other classes. Of note, single agent responses have been observed in lymphomas with TTI-621/TTI-622, which are very rare in CD47 monoclonal antibodies. From a safety perspective, although SIRP-α fusion proteins may have minimal RBC binding and thus less anemia, to date, dose-limiting thrombocytopenia has occurred, which may create challenges with novel combinations with venetoclax or other more myelosuppressive regimens. TTI-622 has planned combination studies with azacitidine for TP53 mutant AML and with azacitidine and venetoclax in TP53 wild-type AML and may induce less thrombocytopenia based on the IgG4 versus IgG1 domain.

Mechanistically, understanding if TP53 mutant patients have distinguishing pathogenic determinants of enhanced sensitivity to CD47 therapies remains a key clinical question. Potentially, the genetic complexity in these patients may increase pro-apoptotic signals, although this remains to be explored. Recent data have also highlighted key immune microenvironment differences in TP53 mutant MDS/AML patients versus wild-type. Specifically, we and others have shown that the leukemia stem cell in these patients has overexpression of PDL1 [46,47]. Additionally, TP53 mutant patients are enriched for an immune suppressive/anergic profile that may be a critical driver for outcomes in this molecular subgroup, although questions remain if perhaps there are differential macrophage populations in this molecular subgroup [47].

## 5. Conclusions

TP53 mutations in MDS/AML patients have emerged as the most negative prognostic factor with inferior outcomes to all standard of care therapies. Most importantly, the data to date strongly support that TP53 mutant patients require investigational therapy in order to improve outcomes. Specifically, TP53 mutant MDS/AML patients have a median OS < 12 months despite standard of care therapy including HMA, HMA + venetoclax, and intensive therapy, as well as very poor outcomes to allogeneic HSCT. However, the treatment landscape for this molecular subgroup is optimistic, with several therapies having synergistic responses, including eprenetapopt and magrolimab (Table 1 and Table 2). As additional myeloid specific therapies are showing robust response rates in MDS/AML patients (e.g., pevonedistat and sabatolimab), understanding how the TP53 mutant subgroup responds is of critical importance given clear differential outcomes in this patient subgroup. Importantly, the durability of responses in this subgroup is of paramount importance and is likely the key endpoint to early phase studies rather than overall response or complete remission rates. Additionally, the TP53 VAF is highly concordant with outcomes and will be a key biomarker in future novel investigations for this subgroup, ideally in the setting of high sensitivity NGS. Likely, we will require a multi-arm attack to truly deliver disease modifying therapy for this subgroup, both pre- and post-HSCT, and perhaps the combination of CD47 directed therapy with p53 reactivation on top of an HMA backbone could prove to be an optimal combination.

## Figures and Tables

**Table 1 ijms-22-10105-t001:** Clinical trials in *TP53* myeloid malignancies.

Treatments (NCI Number)	Phase	Number of Patients	Overall Response Rate	Survival
AZA + VEN NCT04401748	3	431 AML patients total		
		286 (AZA + VEN) (23% *TP53* mutated)	55%	6 months
		145 (AZA + Placebo) (16% *TP53* mutated)	0%	6 months
AZA + Eprenetapopt NCT03072040	1b/2	55 *TP53* mutated MDS/AML patients	71%	10.8 months
AZA + Eprenetapopt NCT03588078	2	52 *TP53* mutated MDS/AML patients	62%	12.1 months
AZA + MAGRO NCT03248479	1b	29 *TP53* mutated AML patients	59%	12.9 months

**Table 2 ijms-22-10105-t002:** Clinical trials ongoing in *TP53* myeloid malignancies.

Treatments (NCI Number)	Phase	Number of Patients	Primary Endpoint
ENHANCE-2 AZA + MAGRO vs. AZA + VEN NCT04778397	3	346 *TP53* mutated AML patients to be randomized	OS
AZA + Placebo vs. AZA + Eprenetapopt NCT03745716	3	154 *TP53* mutated MDS patients included	CR
AZA + VEN + Eprenetapopt NCT04214860	1	51 *TP53* mutated AML patients	Safety
DAC + Cytarabine + ATO NCT03381781	2	100 *TP53* mutated AML patients	RFS
AZA + Eprenetapopt following HSCT NCT03931291	2	33 *TP53* MDS/MAL patients included	RFS

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
