# Peer review of "Personalized Medicine for TP53 Mutated Myelodysplastic Syndromes and Acute Myeloid Leukemia"

_ijms, 2021, doi:10.3390/ijms221810105_

Round 1

Reviewer 1 Report

The paper is an excellent review in the future treatment options for patients with TP53 mutated MDS and AML. It is well written, easy to read, organized. I only have a couple of comments.

  • There is a lot of controversy in the real impact of monoallelic TP53 mutations in outcomes, at least in MDS patients. This is kind of mention during the introduction, but I have missed that you went a little deeper into the subject, as well as that in the conclusion there was a mention that if this is true, the results obtained in the published clinical trials could be affected.
  • In the other hand I have miss a table that resume all the clinical trials mentioned in the review, with phase of the clinical trial, number of patients included, number of patients with TP53 mutations, response date in global and in the TP53 population if available, median survival in global and TP53 population.
  • I will appreciate another table with clinical trials currently recruiting

I have also detected some typos:

  • Line 61: “p<10(-4)”: for me is confuse, maybe better p<0.0001?
  • Line 87: “In TP53 mutated AML patients (n=52)
  • Line 141: “It was significantly related to a lower glomerular filtration rate” instead “It was significantly related to related to a lower glomerular filtration rate”
  • Line 164: “Furthermore, evaluation of novel combinations” instead of “Furthermore, evaluation of novel-novel combinations”

Author Response

Thank you for your comments regarding our manuscript entitled, “Personalized Medicine for TP53 Mutated Myelodysplastic Syndromes and Acute Myeloid Leukemia”, and for providing us sufficient time to address all suggested edits. We hope that the revised manuscript will meet the high standard of quality expected for publication in International Journal of Molecular Sciences. Please find attached our revised manuscript, and below, our point-by-point responses to the reviewers’ comments:

Reviewer 1 : Comments for the Author

The paper is an excellent review in the future treatment options for patients with TP53 mutated MDS and AML. It is well written, easy to read, organized. I only have a couple of comments.

There is a lot of controversy in the real impact of monoallelic TP53 mutations in outcomes, at least in MDS patients. This is kind of mention during the introduction, but I have missed that you went a little deeper into the subject, as well as that in the conclusion there was a mention that if this is true, the results obtained in the published clinical trials could be affected.

We added a discussion of this point in the conclusion.

    In the other hand I have miss a table that resume all the clinical trials mentioned in the review, with phase of the clinical trial, number of patients included, number of patients with TP53 mutations, response date in global and in the TP53 population if available, median survival in global and TP53 population.

We added this table (Table 1).

    I will appreciate another table with clinical trials currently recruiting

We added this table (Table 2).

I have also detected some typos:

    Line 61: “p<10(-4)”: for me is confuse, maybe better p<0.0001?

We did this change.

Line 87: “In TP53 mutated AML patients (n=52)

We did this change.

Line 141: “It was significantly related to a lower glomerular filtration rate” instead “It was significantly related to related to a lower glomerular filtration rate”

We did this correction.

Line 164: “Furthermore, evaluation of novel combinations” instead of “Furthermore, evaluation of novel-novel combinations”

We did this correction.

We hope that you will find this revised manuscript acceptable for a publication in International Journal of Molecular Sciences. Thank you for your consideration.

Sincerely yours,

Thomas Cluzeau and David Sallman

Reviewer 2 Report

General comments:

The article addresses the current therapeutic challenge of the MDS and AML, bearing a TP53 mutation. It is written under a review form, which combined news data regarding the potential treatment strategies. The clonal burden of TP53 was discussed, as well as the hypomethylation drugs venetoclax, eprenapopt or magrolimab in combination with Azacitidine. It is shown that a high VAF and biallelic alterations are high risk, while clearance, even if it is not complete, will lead to a better OS.

The authors make the global statement under the short conclusion paragraph. The article contains too many abbreviations and the presented studies are not explained well enough. Especially the schedule presentations are difficult to read and understand. The written language is poor and article contains many mistakes of syntax and typos. A paragraph regarding types of TP53 disruption in MDS and AML is missing. Background of molecular pathogenesis of TP53 disruption and its role in the MDS/AML outcome is essential for the understanding of the whole therapeutic challenge and possible treatment strategies.

The Concept of the article is very interesting, and this would be a useful text for researchers and clinicians.

Detailed revision and comments:

  1. The structure of the article is not clear- should it be restructured more clearly for example:
  • Introduction
  • Body with more clear structure, to call it “Treatment overview” with
  • Discussion
  • Conclusion
  1. There is also a problem with references that are lacking partially in the introduction or completely in the conclusion. For example: the sentences in the lines 42-50 need respective references. The references listed in the text are generally not readable in the whole article.
  2. The molecular influence of TP53 mutations/ deletions should be explained in a much more detailed way in the introduction. There is a lot of literature on the subject. Why do patients with TP53 mutations have such a bad outcome?
  3. The conclusion is too short and looks rather like an abstract. Should be rewritten with sufficiently presented author’s ideas and corresponding references.

 Detailed revision and comments:

  1. Paragraph : Azacitidine + Venetoclax

Description of molecular mechanism of action of both molecules is missing. The body of this part looks like a short summary from a conference. Should be revised and rewritten.

It is not clear whatever such a big space should be given to the study of Garcia at all (Ref N° 17), since no data were available in the specific TP53 mutated MDS patients? Should all “high-risk” MDS patient being considered as bearing potentially a P53 mutation? If not, why this study is at the really beginning and discussed so in details?

  1. Paragraphs: Azacitidine + Eprenetapopt  and Azacitidine + Magrolimab :

General problems with syntax and data description. The effect of magrolimab should be explained more clearly, as well as the role of calreticulin.

Author Response

Thank you for your comments regarding our manuscript entitled, “Personalized Medicine for TP53 Mutated Myelodysplastic Syndromes and Acute Myeloid Leukemia”, and for providing us sufficient time to address all suggested edits. We hope that the revised manuscript will meet the high standard of quality expected for publication in International Journal of Molecular Sciences. Please find attached our revised manuscript, and below, our point-by-point responses to the reviewers’ comments:

Reviewer 2 : Comments for the Author

The article addresses the current therapeutic challenge of the MDS and AML, bearing a TP53 mutation. It is written under a review form, which combined news data regarding the potential treatment strategies. The clonal burden of TP53 was discussed, as well as the hypomethylation drugs venetoclax, eprenapopt or magrolimab in combination with Azacitidine. It is shown that a high VAF and biallelic alterations are high risk, while clearance, even if it is not complete, will lead to a better OS.

The authors make the global statement under the short conclusion paragraph. The article contains too many abbreviations and the presented studies are not explained well enough. Especially the schedule presentations are difficult to read and understand. The written language is poor and article contains many mistakes of syntax and typos. A paragraph regarding types of TP53 disruption in MDS and AML is missing. Background of molecular pathogenesis of TP53 disruption and its role in the MDS/AML outcome is essential for the understanding of the whole therapeutic challenge and possible treatment strategies.

The Concept of the article is very interesting, and this would be a useful text for researchers and clinicians.

Detailed revision and comments:

The structure of the article is not clear- should it be restructured more clearly for example:

    Introduction

    Body with more clear structure, to call it “Treatment overview” with

    Discussion

    Conclusion

We changed title of each section but included all discussion of each combination in each sections to be more clear. We added some additional discussion in the conclusion.

There is also a problem with references that are lacking partially in the introduction or completely in the conclusion. For example: the sentences in the lines 42-50 need respective references. The references listed in the text are generally not readable in the whole article.

We added and corrected all references to be more readable.

The molecular influence of TP53 mutations/ deletions should be explained in a much more detailed way in the introduction. There is a lot of literature on the subject. Why do patients with TP53 mutations have such a bad outcome?

We added more explanations in the introduction.

The conclusion is too short and looks rather like an abstract. Should be rewritten with sufficiently presented author’s ideas and corresponding references.

Conclusion has been expanded as recommended by the reviewer.

Paragraph : Azacitidine + Venetoclax

Description of molecular mechanism of action of both molecules is missing. The body of this part looks like a short summary from a conference. Should be revised and rewritten.

We added description of molecular MOA of each molecules.

It is not clear whatever such a big space should be given to the study of Garcia at all (Ref N° 17), since no data were available in the specific TP53 mutated MDS patients? Should all “high-risk” MDS patient being considered as bearing potentially a P53 mutation? If not, why this study is at the really beginning and discussed so in details?

We reduced the discussion of this study as suggested by the reviewer.

 Paragraphs: Azacitidine + Eprenetapopt  and Azacitidine + Magrolimab :

General problems with syntax and data description. The effect of magrolimab should be explained more clearly, as well as the role of calreticulin.

We have further refined data description. We have clarified the role of calreticulin and added some additional discussion on the lack of pro-eat me signals on normal cells. We have added some clarity on MOA of magrolimab.

We hope that you will find this revised manuscript acceptable for a publication in International Journal of Molecular Sciences. Thank you for your consideration.

Sincerely yours,

Thomas Cluzeau and David Sallman

Round 2

Reviewer 2 Report

Dear authors, your manuscript is better, but there are still things missing, like for example an explanation about the mechanisms of TP53 . I think that your idea is very interesting, but not yet really worked through. Also, a sound discussion is missing. I suggest that you rethink your approach.

Author Response

Dear Reviewer,

Thank you for your comments regarding our manuscript entitled, “Personalized Medicine for TP53 Mutated Myelodysplastic Syndromes and Acute Myeloid Leukemia”, and for providing us sufficient time to address all suggested edits. We hope that the revised manuscript will meet the high standard of quality expected for publication in International Journal of Molecular Sciences. Please find attached our revised manuscript, and below, our point-by-point responses to your comments:

Reviewer 2: Comments for the Author

Dear authors, your manuscript is better, but there are still things missing, like for example an explanation about the mechanisms of TP53. I think that your idea is very interesting, but not yet really worked through. Also, a sound discussion is missing. I suggest that you rethink your approach.

As requested, we have added additional introduction reviewing the impact of TP53 mutations on a functional level in MDS, particularly the dominant-negative impact of the most common hotspot mutations as described by recent elegant work published in Nature Medicine. We have also added relevance on the specific missense mutation (from the setting of EAp53 score) which together both shed light that therapies that can restore wildtype fx (or are completely p53 independent) are required for activity in this molecular subset. We have also added a comprehensive review on p53 and MDS pathogenesis for the reader as well.   The conclusion is intentionally concise to focus on the treatment landscape of TP53 mutant MDS/AML patients and the key considerations which should be analyzed in prospective studies.

We hope that you will find this revised manuscript acceptable for a publication in International Journal of Molecular Sciences

Sincerely yours,

Thomas Cluzeau and David Sallman

Round 3

Reviewer 2 Report

Dear Authors,

you did a good revisision, and you article is interesting. 

Good luck and best regards